# Effect of Nanoemulsion Containing Enterocin GR17 and Cinnamaldehyde on Microbiological, Physicochemical and Sensory Properties and Shelf Life of Liquid-Smoked Salmon Fillets

**DOI:** 10.3390/foods12010078

**Published:** 2022-12-23

**Authors:** Jiaojiao Duan, Rong Nie, Jing Du, Haoxuan Sun, Guorong Liu

**Affiliations:** 1School of Food and Health, Beijing Technology and Business University, Beijing 100048, China; 2Beijing Engineering and Technology Research Center of Food Additives, Beijing Technology and Business University, Beijing 100048, China

**Keywords:** enterocin Gr17, cinnamaldehyde, nanoemulsion, liquid-smoked salmon fillets, bacteriocin

## Abstract

The spoilage of liquid-smoked salmon represented a serious restriction for shelf life, due to the loss of taste, smell, color and consistency in product quality. The objective of this study was to investigate the feasibility of applying a nanoemulsion delivery system co-encapsulated enterocin Gr17 and essential oils (EOs) to the refrigerated storage of liquid-smoked salmon. The synergistic inhibiting effects of enterocin Gr17 and EOs were evaluated, a nanoemulsion delivery system with the optimal combination was developed, and the evolution of the microbiological, physicochemical, and sensory properties of liquid-smoked salmon fillets were analyzed during a 49-day period of refrigerated storage. The results showed that the combination of enterocin Gr17 and cinnamaldehyde essential oil (CEO) displayed the strongest synergistic inhibiting effect on foodborne pathogens. A nanoemulsion system incorporating enterocin Gr17 and CEO was successfully developed and presented a broad spectrum of activity against most of the tested bacteria. A nanoemulsion system incorporating enterocin Gr17 and CEO (CO-NE) could significantly inhibit the growth of microflora, suppress the accumulation of total volatile basic nitrogen (TVB-N) and thiobarbituric acid reactive substance (TBARS), and maintain better color, texture, and sensory profiles during smoked salmon storage at 4 °C. Overall, from a microbiological, physicochemical, and sensory point of view, the CO-NE treatment could extend the shelf life to 42 days and maintain the relatively low TVB-N value (≤15.38 mg/100 g), TBARS value (≤2.51 mg MDA/kg), as well as a relatively high sensory score (≥5.83) during the whole storage period. Hence, a nanoemulsion system incorporating enterocin Gr17 and CEO could be a promising bio-preservative technology and alternative to the conventional processes used for improving the safety and quality of chilled liquid-smoked salmon.

## 1. Introduction 

Liquid-smoked salmon is a crucial resource that contains high-quality dietary fat, proteins, vitamins, and minerals. It is popular around the world because of its rich nutritional value and distinct smoky flavor [1,2]. However, the abundance of fat (8–14%) and protein (20–22%) in liquid-smoked salmon makes it particularly sensitive to rapid microbial growth and lipid oxidation, which, in turn, poses serious difficulties during storage and transportation [3]. Therefore, preservatives must be added at proper concentrations to guarantee the safety, nutrition, and sensory properties of liquid-smoked salmon fillets. Owing to the demand for natural, environmentally friendly, and healthy food from consumers, natural bioactive agents, have significant advantages compared to traditional chemical preservatives [4].

Bacteriocins are proteins or peptides that exhibit antibacterial activity against various spoilage and pathogenic bacteria [5]. Bacteriocins from lactic acid bacteria (LAB) are non-toxic, highly potent, and safe, and so have been widely used as preservatives for food products [6]. However, bacteriocins have a relatively narrow spectrum, exerting strong inhibitory effects against Gram-positive (G^+^) bacteria and weak and even non-inhibitory effects against Gram-negative (G^−^) bacteria [7]. Essential oils (EOs) are complex mixtures of volatile organic compounds produced by plants, and they frequently exhibit antimicrobial activity against G^+^ and G^−^ bacteria as well as antioxidant properties. A high level of EOs in food showed unacceptable organoleptic properties, and low levels of EOs could not exert antibacterial effects in food products [7,8]. The combination of bacteriocin and EOs demonstrates a synergistic effect and broadens the antibacterial spectrum of bacteriocins, decreasing the usage of both components, thereby effectively maintaining or improving the sensory attributes, inhibiting microbial growth, and delaying chemical changes to food during storage [5,6,7,8]. In addition, the cost of additives would be further reduced using a combination of bacteriocins and EOs, facilitating their commercial production [9]. Ghrairi et al. [10] found that the combination of enterocin with EOs induced a synergistic effect against *L. monocytogenes* and *E. coli*. A mixed natural preservative presented better antimicrobial effects in milk with a combination of 16.25 μg/mL of cinnamaldehyde and 31.25 μg/mL of nisin, reducing the usage by about 75% compared with the individual MIC for cinnamaldehyde and nisin, respectively [11]. 

However, there are several challenges regarding the use of bacteriocins and EOs in actual food products. The unwanted interactions between food matrixes and bacteriocins, degradation caused by enzymes in the food system, and varied pH all decreased the antibacterial activity of bacteriocins [11,12]. Bacteriocins with a high content of hydrophobic amino acids are easily bound to charged or hydrophobic macromolecules in food products [13]. The fat content of smoked salmon is about 8~14%, decreasing the antimicrobial activity of bacteriocins. As for EOs, intrinsic obstacles, such as their low water solubility, high volatility, low stability, and negative organoleptic effects, affect their antibacterial and antioxidant activities in food systems [14]. Instead of applying bacteriocins and bulk EOs to foods directly, one promising strategy is to encapsulate bacteriocins and EOs into a nanoemulsion, which can be used as a colloidal delivery system to protect the effects of bioactive compounds [15]. Moreover, the sustained release of a nanoemulsion enables a longer action time for the active substances, maintaining their antimicrobial activity for longer periods [16]. A nanoemulsion incorporating star anise essential oil and nisin is not only beneficial for the shelf-life extension of chicken fillets but also has no adverse effects on the flavor of chicken [17]. Additionally, star anise essential oil, polylysine, and nisin-loaded nanoemulsions revealed efficient activities when applied to ready-to-eat (RTE) pork meat products as antimicrobial agents [18]. However, the application of bacteriocin-EO-loaded nanoemulsion delivery systems to RTE seafood products to inhibit the spoilage process requires further study. 

Enterocin Gr17 (ENT) is a previously discovered broad-spectrum bacteriocin originating from *Enterococcus faecalis* Gr17 that is conserved in our laboratory [6]. It demonstrates extensive inhibitory effects on many foodborne pathogens (*Listeria monocytogenes*, *Staphylococcus aureus*, *Bacillus subtilis*, and *Escherichia coli*, etc.) and has the potential use as a natural additive in liquid-smoked fish storage [6]. Thus, the objective of this study was to investigate the feasibility of an enterocin Gr17 and EO-loaded nanoemulsion delivery system application in liquid-smoked salmon storage. Specifically, the optimal combination of enterocin Gr17 and the EOs was screened using the broth microdilution checkerboard method, then the physicochemical properties and the antimicrobial activity of the enterocin Gr17–cinnamaldehyde-loaded nanoemulsion were studied to develop an efficient and stable nanoemulsion. In addition, the effects of the nanoemulsion treatments on the microbial properties (total viable counts and specific spoilage counts), physicochemical characteristics (active thiobarbituric acid and total volatile nitrogen), and sensory quality (color, texture, and flavor components) of vacuum-packed liquid-smoke salmon fillets at 4 °C were also assessed. 

## 2. Materials and Methods

### 2.1. Bacterial/Fungal Strains and Chemicals 

The enterocin producer of *E. faecalis* Gr17, which was isolated from a traditional Chinese low-salt fermented whole-fish product, was incubated in Man, Rogosa and Sharpe (MRS) broth (LuQiao, Beijing, China) at 37 °C. *Staphylococcus aureus* (ATCC 6538), *Bacillus cereus* (ATCC 11778), *Listeria monocytogenes* (ATCC 35152), *Salmonella enterica* (CGMCC 1.10603), *Escherichia coli* (CGMCC 1.90), *Pseudomonas aeruginosa* (ATCC 9027), *Pseudomonas fluorescens* (ATCC 14150), *Shewanelle putrefaciens* (ATCC 8071), *Canidia Albicans* (ATCC10231), *Aspergillus flavus* (CGMCC 3.15726), *Penicillium expansum* (CGMCC 15686), and *Alternaria alternata* (CFCC 87705) were used as the indicator strains for the antimicrobial activity measurements. *S. aureus*, *B. cereus*, *L. monocytogenes*, *S. enterica*, *E. coli*, *P. aeruginosa*, *P. fluorescens*, and *S. putrefaciens* were grown in trypticase soy broth (TSB), and *C. albicans*, *A. flavus*, *P. expansum*, and *A. alternata* were grown in potato dextrose broth (PDB). TSB and PDB were purchased from LuQiao Institute (Beijing, China). 

The essential oils (EOs) (purity: 98.0%), including eugenol (EEO), thymol (TEO), carvacrol (CAEO), cinnamaldehyde (CEO), menthone (MEO), and linalool (LEO), were purchased from Macklin Institute (Shanghai, China). The emulsifiers (food-grade), including the soybean protein isolate (SPI, >90% protein), whey protein isolate (WPI, 97.7% protein dry basis), sodium caseinate (SCN, purity: 90.0%), hydroxypropyl methylcellulose (HPMC, viscosity 4000–6500 mPas, 2% in distilled water at 20 °C), konjac gum (KGM, viscosity ≥ 15,000 mPas, 1% in distilled water at 20 °C), and xanthan gum (XGM, viscosity ≥ 1200 cps, 1% in KC1 at 20 °C), were purchased from Sigma-Aldrich Company (St Louis, MO, USA). The 2,2-diphenyl-1-picrylhydrazyl (DPPH) measurement kits, the thiobarbituric acid reactive substance (TBARS) assay kits, and the total volatile basic nitrogen (TVB-N) assay kits were purchased from the Jiancheng Bioengineering Institute (Nanjing, China). Fresh salmon was bought from a local market in Beijing, China.

### 2.2. Preparation of Partially Purified Enterocin Gr17 

Enterocin Gr17 (ENT) was purified from the supernatant of *E. faecalis* Gr17 using a four-step procedure consisting of ammonium sulfate precipitation, SP-Sepharose fast-flow cation exchange, Sephadex G25 gel filter chromatography, and reverse-phase high-performance liquid chromatography, as described previously [6]. Approximately 1.67 mg of partially purified ENT (purity: 87.4%) was obtained from each 1000 mL culture of *E. faecalis* Gr17.

### 2.3. Synergistic Antimicrobial Effect of Enterocin Gr17 and EOs

The minimum inhibitory concentrations (MIC) of enterocin Gr17 (ENT) and EOs against the indicator strains were determined using the broth microdilution method in 96-well microplates, according to Buldain et al. [19]. Each of the wells was filled with 100 μL of microbial suspensions, with an inoculum concentration of 10^6^ CFU/mL. Then, serial dilutions of ENT and EOs in TSB/PDB medium were added to the wells, with the final concentrations of 20, 10, 5, 2.5, 1.25, 0.625, 0.3125, 0.1563, 0.0781, 0.0390, and 0.0195 mg/mL. The wells that only had microbial suspensions (10^6^ CFU/mL) were used as a negative control. The microplates of pathogens (*S. aureus*, *B. cereus*, *L. monocytogenes*, *S. enterica*, and *E. coli*) were incubated at 37 °C for 24 h; the spoilage microorganism (*P. aeruginosa*, *P. fluorescens*, and *S. putrefaciens*) were cultivated at 30 °C; and the plates of fungi (*C. albicans*, *A. flavus*, *P. expansum*, and *A. alternata*) were incubated at 25 °C for 48 h. After cultivation, the mixtures in the 96-well microplates were measured at 600 nm. The MIC was the concentration of antimicrobial agent causing a 50% growth inhibition for the indicator strains, and it is expressed by mg/mL. 

The synergistic antimicrobial effect of ENT and the EOs was studied using the broth microdilution checkerboard method [19]. Serial dilutions of both antimicrobials were mixed to obtain a fixed amount of the first agent and increasing amounts of the second agent in each row (and column). Eight two-fold serial dilutions (from 2MIC to MIC/64) of ENT and the EOs were prepared. The 50 μL dilutions with different concentrations of ENT and EOs were added to the vertical and horizontal orientations of the 96-well microplates, respectively. A total of 100 μL of fresh microbial suspension (10^6^ CFU/mL) was added to each well and cultured, as described above. The combined effects were evaluated in terms of the fractional inhibitory concentration index (FICI). The FICI was calculated as follows:(1)$${\mathrm{FICI}=(A)/(\mathrm{MIC})}_{a}{+(B)/(\mathrm{MIC})}_{b}\;$$ (A): MIC of ENT in combination with EO; (B): MIC of EO in combination with ENT; (MIC)_a_ and (MIC)_b_: MIC of the ENT and EO alone, respectively. The FICI was interpreted as follows: ≤0.5: synergistic activity, 0.5–1: additive activity, 1–2: indifference, >2: antagonism.

### 2.4. Preparation of the Nanoemulsions

The nanoemulsions (NEs) were formulated according to the method of Gahruie et al. [20], with subtle alterations. Six types of emulsifiers, including SPI, WPI, SCN, HPMC, KGM, and XGM, were used to prepare the different nanoemulsions. Firstly, different types of emulsifiers (SPI [2% *w*/*w*], WPI [2% *w*/*w*], SCN [6% *w*/*w*], HPMC [1% *w*/*w*], KGM [0.2% *w*/*w*], and XGM [0.2% *w*/*w*]) were dissolved in distilled water and stirred overnight at ambient temperature. Then, ENT was mixed with the aqueous phase at a final concentration of 1% (*w*/*w*). Then, EO (10% *w*/*w*) was added to the above solution and stirred slowly at a ratio of 1:9 (*w*/*w*). The primary emulsion was prepared with high-speed shear apparatus (XHF-DY, Ningbo Scientz Biotechnology, Ningbo, China). Then, the nanoemulsion was produced using a homogenizing ultrasonic homogenizer (SONOPULS, HD3200, 20 kHz, 150 W, BANDELIN Co., Berlin, Germany). The NEs were kept at 4 °C for further analysis.

### 2.5. Physicochemical Characteristics of Nanoemulsions

#### 2.5.1. Particle Size and Zeta Potential

The particle size, polydispersity index (PDI), and zeta potential of the NEs were examined using a dynamic light-scattering instrument (Zetasizer Nano ZS, Malvern Instruments, Malvern, UK), according to the method described by Sui et al. [21]. The freshly prepared NE samples were diluted (1:100) using ultrapure water before the measurement was taken. The settings for the analyzer were as follows: flow rate: 50%; run-time: 30 s; temperature: 25 °C; relative refractive index: 1.523; zeta potential range: −100 to 50 mV.

#### 2.5.2. Stability Analysis

The stability of the NE samples was estimated using the Turbiscan Lab Expert (Formulaction, Toulouse, France) [22]. After being added to the scanning tube, 20 mL of the emulsion sample was scanned at 25 °C for 12 h every 30 min. The scan results were expressed as the Turbiscan stability index (TSI).

#### 2.5.3. Encapsulation Efficiency 

The encapsulation efficiency (EE) of the NEs was measured using the method of Singh et al. [23]. The NE samples were centrifuged under the following conditions: 10,000 g for 10 min at 4 °C. The concentration of ENT in the supernatant was estimated using a BCA Protein Assay Kit (Beyotime, Shanghai, China). A UV spectrophotometer (UV-3600Plus, Shimadzu, Japan) was used to determine the concentration of CEO. The encapsulation efficiency (*EE*) was calculated according to the following formula:(2)$$EE=U_{0}-U_{1}/U_{0}\times 100\%$$
where *U*_0_ is the total additional mass of ENT or EO and *U*_1_ is the mass of free ENT or EO in the fresh nanoemulsion samples.

#### 2.5.4. Shear Viscosity Measurement

The shear viscosities of the nanoemulsion samples were determined using a dynamic shear rheometer (Kinexus, Malvern, Worcestershire, UK) with a 40 mm rheometer plate at 25 °C and the measured gap was 1 mm [24]. The NE samples were equilibrated on the rheometer plate for 5 min prior to the measurement. The viscosity values obtained at the shear rates between 0.1 and 200/s were recorded to describe the rheological parameters of the NE samples.

### 2.6. Antibacterial and Antioxidant Activity of Nanoemulsions

The antibacterial activity of the NE samples against the indicator strains mentioned above was analyzed using the agar diffusion method, as described by Abdollahi et al. [25], with some modifications. Briefly, 0.2 mL of the NE sample was added into the wells (depth: 5 mm; diameter: 9 mm) in an agar plate with 10^6^ CFU/mL microorganisms [26]. Meanwhile, a pure TSB/PDB without any sample was used as a negative control, and the solutions of ENT and the EOs alone at the same concentration were used as the positive controls. The plates were incubated as described in Section 2.3. The antibacterial activity was evaluated by measuring the diameter of the inhibition zone, which was measured as the middle line starting from the clear spot around the well. An inhibition zone diameter lower than 10 mm was classified as having no inhibition, an inhibition zone diameter between 10–20 mm was considered to have a normal inhibition, and an inhibition zone diameter greater than 20 mm was defined as a strong inhibition [27,28]. 

The antioxidant activity of the NE samples was evaluated using the DPPH method mentioned by Almasi et al. [29] with minor modifications. Briefly, 0.5 mL of the sample supernatant was prepared by centrifugation at 5000 rpm for 10 min and mixed with 4.5 mL of methanolic DPPH solution (1 mmol/L). After being vortexed, the mixture was kept in the dark for 30 min so that the reaction could be carried out. The absorbance of all samples was detected at 517 nm.
(3)$${Antioxidant\;activity=A}_{1}{-A}_{2}{/A}_{1}\times 100\%$$
where *A*_1_: the absorbance of the DPPH blank solution; *A*_2_: the absorbance of the DPPH and sample solution.

### 2.7. Effects of Nanoemulsions System Application on the Quality Characteristics of Liquid-Smoked Salmon Fillets

#### 2.7.1. Treatment of Liquid-Smoked Salmon Fillets Samples

The salmon was sliced into skinless fillets in a sterile environment and rinsed with sterile water. After being dry-salted with fine-refined salt at 4 °C for 12 h, all of the fillets were dipped in a 1.5% (*v/v*) liquid-smoke solution for 15 minutes. The SPI-NE only loaded with ENT (ENT-NE) or CEO (CEO-NE) were prepared as in *2.4*. The fillets (25 g) were then immersed for 15 min in plates added with 100 mL ENT-NE (0.9% *w*/*w*), CEO-NE (1% *w*/*w*) and SPI-ENT/CEO-NE (CO-NE) (ENT 0.9% [*w*/*w*]; CEO 1% [*w*/*w*]), respectively. The untreated salmon fillets were used as the control samples. A 10-liter Enviro-Pak CHU-150 commercial smoker (Clackamas, OR, USA) was used for 60 min at 30 °C. All of the samples were vacuum-packed in sterile polyethylene bags and stored in a refrigerator at 4 °C for up to 49 days. The fish fillet samples treated using the nanoemulsion without ENT and CEO incorporation were used as a control. The liquid-smoked salmon fillets were randomly selected from each group for analysis every seven days.

#### 2.7.2. Physicochemical Parameters

The physicochemical properties analyzed included TBARS, TVB-N, color, and texture. TBARS was measured using a modified version of the method described by Guo et al. [30]. The samples (10 g) were homogenized with trichloroacetic acid (20 mL, 5% [*w*/*v*]) and butylated hydroxytoluene (500 mL, 10 mg/ml) and filtered. The supernatant was mixed with 5 mL of TBA reagent and heated at 94 °C for 20 min before cooling. The absorbance was measured at 532 nm. The result was expressed as mg Malondialdehyde (MDA)/kg. 

The TVB-N of the liquid-smoked salmon fillets was estimated according to the method of Uriarte-Montoya [31]. The TVB-N values were expressed as mg N/100 g. The quantification limits of TBARS and TVB-N were 3 mg MDA/kg and 30 mg N/100 g, respectively [32,33].

The instrumental color (CIE *L**, *a**, *b**) of the liquid-smoke salmon samples was measured using a Minolta Chromo meter (CM-3610A, Tokyo, Japan) with the standard illuminant D65, calibrated by a white reference (CR-A44). The samples at three equidistant points were determined for *L**(lightness), *a** (redness), and *b** (yellowness) indicators. 

The texture of the liquid-smoke salmon samples was determined using a TA XT2 Texture Analyzer (Stable Micro Systems, Godalming, UK. In addition, the test conditions of the texture profile analysis (TPA) were as follows: a compression cycle of 30% twice consecutive and a cycle interval of 5 s. The probe operated at a speed of 3 mm/SEC. The data analysis was performed by using texture exponent software, and the following parameters were calculated: hardness (N), adhesiveness (mJ), springiness(mm), and chewiness (mJ).

#### 2.7.3. Microbiological Analysis

To assess bacterial contamination and degree of spoilage for liquid-smoked salmon stored at 4 °C, the total viable counts (TVC), psychrophilic bacteria counts (PBC), total *Pseudomonas* counts (TPC) and H_2_S-producing bacteria counts (HBC) were detected. The liquid-smoked salmon samples (25 g) were homogenized with 0.9% (*w*/*v*) sterile NaCl solution (225 mL) for 1 min in a Scientz-11 Blender (Seward Medical Ltd., London, UK), and then the suspension was diluted 10 × continuously with sterile water. The TVC and PBC were incubated in plate count agar (LuQiao, Beijing, China) at 30 °C for 3 days and 10 °C for 7 days, respectively. The TPC and HBC were cultured on an iron agar medium and CFC-selective medium, respectively. The results were expressed as log_10_ CFU/g. The quantification limit was set at 10^6^ CFU/g for TVC, TPC, PBC, and HBC [33,34,35].

#### 2.7.4. Volatile Compounds Analysis

The volatile compounds in the liquid-smoked salmon fillets on 0, 21st, and 49th day were detected, as mentioned by Guo et al. [36], with minor modifications. The volatile compounds in the salmon fillets, as obtained by SPME, were separated and identified using gas chromatography-mass spectrometry (GC-MS) analysis (Finnigan Trace GC-MS, Thermo Fisher, USA). A 2 g mass of the salmon sample, 1 g NaCl, 5 mL of ultrapure water and 10 uL of internal standard solution 6-undecanone (1000 ug/L in methanol) were placed into a 20 mL sampling vial, which was placed in a water bath for 20 min at 60 °C for equilibration. A 50/30 μm DVB/CAR/PDMS (Divinylbenzene/Carboxen/Polydimethylsiloxane) fiber was selected because of its high sensitivity for polar compounds, non-polar compounds, and small molecule compounds. The SPME fiber was inserted into the headspace of the glass vial about 1 cm above the sample surface for 40 min at 60 °C. After extraction, the fiber was inserted into the injection port of a GC (250 °C) for 3 min to desorb the analytes. The DB-5 MS capillary column (30 m × 0.25 mm × 0.25 μm, J&W Scientific Inc., Folsom, CA, USA) was used under the following conditions: oven temperature was maintained at 40 °C for 3 min and programmed to 100 °C at a rate of 3 °C/min, then programmed to 230 °C at a rate of 5 °C/min, and then programmed to 280 °C at a rate of 15 °C/min and kept for 10 min. Besides, the parameters of the mass spectrometer were as follows: splitless injection technique, Electron Ionization (EI) mode, ionization energy 70eV, scan mass range of 33–500 m/z, interface line temperature 280 °C and ion source temperature 220 °C. Use 6-undecanone was used as an internal standard for quantitative analysis. The odor activity value (OAV) of a compound was calculated by dividing the calculated concentrations by the sensory thresholds in the literature, as obtained from the literature.

#### 2.7.5. Sensory Analysis

The sensorial evaluation of the liquid-smoked salmon fillets was conducted using the quantitative descriptive analysis method [37]. The sensory attributes (color, odor, taste, texture, and overall acceptance) of the liquid-smoke salmon fillets were evaluated by 10 trained panelists. The intensities of each sensory attribute were scored using a ten-point scale: 0 = none or having no perceptible intensity; 3 = weak intensity; 5 = moderate intensity; 7 = high intensity; and 10 = extremely high intensity. The panel members were asked to state whether the smoked salmon was acceptable or not, and a sensory score of 5 was used as the limit of acceptability for determining the shelf life of smoked salmon. The evaluation was performed every 7 days for the fillets stored at 4 °C during storage from 0 to 49 days. Each sample was evaluated three times by each panelist. The final sensory score of each attribute was expressed as a mean and was used to plot a radar diagram.

### 2.8. Statistical Analysis

All of the experiments were repeated in triplicate. The results were reported as mean values ± standard deviation. For the data of physicochemical characteristics and antibacterial and antioxidant activity of nanoemulsions, the means were compared by one-way ANOVA test (*p* < 0.05) analysis using SPSS 23.0 (SPSS Inc., Chicago, IL, USA) and MetaboAnalyst 5.0. For liquid-smoked salmon, results of the microbiological, physicochemical and sensory properties were treated by two-way ANOVA test (*p* < 0.05) analysis to test the effect of nanoemulsions system application with SPSS 23.0 and MetaboAnalyst 5.0. The identification of the volatile compounds in the smoked salmon fillets was achieved by using Xcalibur 4.2 (Thermo Fisher Scientific, Waltham, MA, USA). 

## 3. Results

### 3.1. Antimicrobial Effect of Enterocin Gr17 and EOs Combinations on Bacterial and Fungal Strains 

The MIC and FICI of ENT and the EOs against bacterial and fungal strains are shown in Table 1. The MIC of ENT was higher than that of most EOs when they were against the same strain, which indicated that the inhibition of ENT on the indicator strains was weaker than most EOs. Among the EOs single treatment, CEO had the most obvious antimicrobial effect on most bacterial and fungal strains, excluding *P. expansum*. The inhibitory effect of CEO and TEO on *C. albicans* was significantly higher than that of the other EOs (*p* < 0.05), with MICs of 0.0195 g/L and 0.1563 g/L, respectively. Meanwhile, an additive or indifferent effect was observed in most combinations of ENT and EOs, and no antagonism was found in the combinations (Table 1). The fungi exhibited decreased sensitivity compared to the bacterial strains under the treatment of ENT and EOs, while the combination of ENT and CEO had the best synergistic effect, which was effective against all of the indicator strains, including bacteria and fungi. Therefore, the combination of ENT and CEO was selected to conduct the subsequent study.

### 3.2. Effect of Emulsifier Type on Properties of Nanoemulsions

Droplet size, zeta potential, PDI, and TSI are considered essential parameters to demonstrate the stability and efficacy of NE. As shown in Table 2 and Appendix A, the type of emulsifier significantly affected the size, zeta potential, PDI, and TSI of the NE. The SPI-stabilized NE containing ENT and CEO (CO-NE) had the smallest average droplet size of 161.26 ± 6.40 nm and zeta potential of 32.51 ± 0.83 mV. The absolute values of the potential values larger than 25 mV indicate that the emulsion system is stable, which could be associated with the strong electrostatic repulsive force between the droplet particles. The force could be considered a barrier, which possibly prevents droplet aggregation, leading to an increase in droplet size. Compared with droplet size and zeta potential, PDI can better reflect the stability and sustained release of emulsion. PDI values more than 0.7 indicate large particle size distribution and good stability. With SPI as the emulsifier, the CO-NE exhibited minimized PDI (0.235) and TSI (2.50 ± 0.06) compared to the NE stabilized by the other emulsifier, indicating the best dispersity and optimal stability. Furthermore, the EE values of NEs with different emulsifiers are represented in Figure 1. The highest EE was observed in the CO-NE, both for ENT (96.62 ± 0.53%) and CEO (96.94 ± 1.06%). The results suggested that SPI was the most suitable emulsifier to prepare the NE containing ENT and CEO.

### 3.3. Effects of Nanoemulsions System Applied on Liquid-Smoked Salmon Fillets 

Viscosity is considered to be an important indicator to show the properties of NE, indirectly reflecting the physical stability of NE. The blank SPI solution was used as a control. The viscosity versus and shear rate curves revealed that all of the NE samples exhibited non-Newtonian shear-thinning behavior, in which the viscosity decreased with the increasing shear rate (Figure 2). After the addition of ENT and CEO (single or mixed), the viscosity of SPI decreased significantly, and the viscosity of CO-NE was lowest among all of the treatments. The single or mixed addition of ENT and CEO led to an interfacial rheology variation in the NE, then unstable interfacial layers formed, which caused a lower viscosity. Moreover, the formation of the unstable interfacial layer may be due to the accumulation of soybean protein on the droplet surface, which decreases the effective content of soybean protein in the solution. The lower viscosity of CO-NE may be caused by the fact that more soy protein was adsorbed to the droplet surface due to the presence of more ENT and CEO in the NE system. The release rate of ENT and CEO across 50 h was tested, and the release rate of ENT and CEO in the SPI-based NE revealed a continuously increasing trend across 0–50 h, reaching a maximum of 94.11 ± 4.62 % and 96.30 ± 5.01%, respectively. These results demonstrated that a significant, sustained release effect was achieved by containing the ENT and CEO within an SPI-based NE (Appendix A).

### 3.4. Antimicrobial and Antioxidant Activity of the SPI-Based Nanoemulsions

To evaluate the antimicrobial activity of the SPI-based NEs, the inhibitive effects of the NEs on bacterial and fungal strains were tested (Table 3). The results showed that CEO-NE had stronger inhibitory effects than ENT-NE on most bacteria and fungi, excluding strains of *P. fluorescens* and *S. putrefaciens*. CO-NE had a stronger antibacterial activity against all of the tested strains than that in ENT and CEO single-encapsulated NEs. In addition, *S. aureus* showed the most sensitivity to CO-NE; the inhibitory zone was 42.4 ± 0.31 mm.

The DPPH-scavenging activity of the NE samples is shown in Figure 3. CO-NE had the maximum DPPH-scavenging activity of 73.5 ± 0.71%, and the DPPH scavenging activity of CEO-NE (58.7 ± 0.64%) was significantly higher than ENT-NE (34.0 ± 0.27%). The higher DPPH-scavenging activity of CEO-NE may be due to the differences in the antioxidant activity between ENT and CEO.

### 3.5. Effect of Nanoemulsions on the Quality Caracteristics of Liquid-Smoked Salmon Fillets

#### 3.5.1. Microbiological Parameters

The TVC, TPC, PBC, and HBC among all of the treatments were determined during refrigerated storage at 4 °C. Different NEs reduced the number of microorganisms in the fish by varying degrees. As shown in Figure 4a, the TVCs of all treatments increased with the increasing storage time. The TVCs were initially 2.26 log_10_ CFU/g for all of the treatments. After 4 °C storage for 49 days, the TVC reached 10.86, 9.15, 9.09, and 6.28 log_10_ CFU/g for the control, ENT-NE, CEO-NE, and CO-NE groups, respectively. Compared with the control, the TVCs of the NE treatments (ENT-NE, CEO-NE, and CO-NE) significantly decreased by 1.71, 1.77, and 4.58 log_10_ CFU/g, respectively. The TVCs of the control (6.66 ± 0.17 log_10_ CFU/g) exceeded the limit on the 28th day of storage, and the ENT-NE-treated samples (6.26 ± 0.25 log_10_ CFU/g) and the CEO-NE-treated samples (6.31 ± 0.21 log_10_ CFU/g) exceeded the limit (6 log_10_ CFU/g) on the 35th day of storage. However, the TVC values of the CO-NE-treated fillets were 4.59 ± 0.15 log_10_ CFU/g on the 28th day, 4.97 ± 0.15 log_10_ CFU/g on the 35th day, and remained over the limit until the 49th day. Therefore, the smoked salmon control, ENT-NE, CEO-NE, and CO-NE remained within the microbiological limits for up to 21, 28, 28, and 42 days of storage at 4 °C, respectively. Compared with single ENT-NE and CEO-NE, CO-NE could more effectively reduce the TVC of the salmon fillet. 

Similar growth trends were observed for TPC, PBC, and HBC during storage (Figure 4b–d). The TPC, PBC, and HBC in the control reached more than 6 log_10_ CFU/g on the 28th, 35th, and 35th day, respectively. The TPC, PBC, and HBC of both the ENT-NE-treated and CEO-NE-treated fillets exceeded 6 log_10_ CFU/g from the 35th, 42nd, and 42nd day, respectively. However, the TPC of the CO-NE-treated fillets increased to above the minimum acceptable level until the 49th day, and the PBC and HBC in CO-NE did not reach the acceptable limit until the 49th day. Moreover, the TPC, PBC, and HBC of the ENT-NE-, CEO-NE-, and CO-NE-treated fillets decreased by 1.71–4.46, 1.40–4.43, and 1.34–4.48 log_10_ CFU/g on the 49th day, respectively, as compared to the control. CO-NE showed the lowest TPC, PBC, and HBC on the 49th day, demonstrating the best antimicrobial effects among the three NEs, and the antimicrobial effect of ENT-NE and CEO-NE was not significantly different (*p* > 0.05). 

#### 3.5.2. Lipid Oxidation and Protein Degradation

The TBARS values of the salmon fillets under different NE treatments are displayed in Figure 5a. The TBARS index of all groups increased with increasing storage time. The initial TBARS value was 1.18 mg MDA/kg and reached maximum values of 4.38, 3.83, 3.78, and 2.51 mg MDA/kg in the control, ENT-NE, CEO-NE, and CO-NE group on the 49th day, respectively. Compared with the control, the TBARS of the ENT-NE, CEO-NE, and CO-NE groups decreased significantly (*p* < 0.05) by 22.43%, 25.82%, and 55.89% on the 42nd day, respectively. The TBARS index of the control, ENT-NE, and CEO-NE exceeded 3 mg MDA/kg on the 35th, 42nd, and 42nd days, respectively. However, the TBARS of CO-NE was not beyond the acceptable limit throughout the 49 days. The results indicated that CO-NE exhibited the best ability to inhibit lipid oxidation, which may be attributed to the synergistic antioxidant activity of ENT and CEO. Noticeably, there was no significant difference between the TBARS of the ENT-NE and CEO-NE groups. 

The TVB-N values of the salmon fillets under different NE treatments are shown in Figure 5b. A higher TVB-N value means a higher degree in the breakdown of proteins and other nonprotein nitrogen-containing compounds, such as nucleic acids, indicating the worse quality of the product. The TVB-N of the fillets in the control increased from 6.45 ± 0.41 mg/100 g to 30.11 ± 0.71 mg/100 g across 0–49 days. The TVB-N index of the control exceeded 30 mg/100 g on the 49th day. However, the TVB-N of ENT-NE, CEO-NE, and CO-NE did not exceed the acceptable limit throughout the 49 days. Compared with the control group, the increase in TVB-N was more gradual in the experimental groups, with 22.03 ± 1.22, 22.15 ± 1.60, and 15.38 ± 0.98 mg/100 g observed on the 49th day for ENT-NE, CEO-NE, and CO-NE, respectively. Compared with the control, the TVB-N of the ENT-NE, CEO-NE, and CO-NE groups decreased significantly (*p* < 0.05) by 34.27%,33.69%, and 62.17% on the 49th day, respectively. The results showed that the inhibitory effect of CO-NE was more efficient than ENT-NE and CEO-NE in the proteolysis of smoked salmon. 

#### 3.5.3. Color and Texture Properties

The color changes in the liquid-smoked salmon fillets in the different groups are shown in Table 4. Initially, there was no significant difference (*p* > 0.05) in the control and treatments on the *L**, *a**, and *b** values at 0 days, indicating that the NEs had no obvious effect on the color of the salmon fillets. The decrease in the *L** value means that the color of the fillets was darker, and the gloss was reduced. The *L** value of the control decreased significantly (*p* < 0.05) by 15.51 ± 0.38 between 0 and 49 days; however, a slight decline in the *L** value was observed in the ENT-NE, CEO-NE, and CO-NE treatments by 8.43 ± 0.28, 8.28 ± 0.18, and 1.07 ± 2.60 between 0 and 49 days, respectively. The results indicated that the three NEs could restrain the decreasing lightness of the fillets during storage and that CO-NE was more effective than ENT-NE and CEO-NE. Compared with the *a** value of 11.65 ± 0.95 in the control, higher *a** values for the ENT-NE-, CEO-NE-, and CO-NE-treated fillets were observed: 15.54 ± 1.96, 17.04 ± 1.72, and 21.14 ± 1.28 on the 49th day. The results showed that NEs could help to maintain good redness and that CO-NE exhibited the best retention effect. Regarding parameter *b**, it had no significant change across all groups, as, during the increased storage, no obvious changes in the yellowness of the fillets were revealed. In general, CO-NE was more effective than ENT-NE and CEO-NE in restraining the decreasing lightness and redness. The preventative effect might be attributed to the antibacterial and antioxidant activity of the active ingredients contained in the NE and the barrier capacity of the NE.

The results of the texture analysis are shown in Figure 6. In all groups, decreased hardness, adhesiveness, springiness, and chewiness of the salmon fillets were observed during storage. The NEs could maintain the texture of the salmon fillets to different degrees. For the hardness value, it significantly (*p* < 0.05) decreased on the 21st day in the control samples. Compared to the control, the hardness of the ENT-NE-, CEO-NE-, and CO-NE-treated fillets began to decrease on the 21st, 28th, and 42nd day, respectively. The reduced hardness of the control, ENT-NE, CEO-NE, and CO-NE treatments were 6.30 ± 1.30, 3.70 ± 0.24, 4.11 ± 1.64, and 2.45 ± 0.92 N across the 0–49 days, respectively, confirming the protective effect of NEs regarding hardness. These data reflected that CO-NE could more effectively prevent reduced hardness. The analysis of the adhesiveness, springiness, and chewiness also produced similar results to the analysis for hardness. During the 49 days of storage, the adhesiveness, springiness, and chewiness of the control samples decreased by 0.34 ± 0.05 mJ, 0.83 ± 0.14 mm, and 2.03 ± 0.16 mJ, respectively. However, the ENT-NE-, CEO-NE-, and CO-NE-treated fillets only decreased by 0.15 ± 0.04, 0.13 ± 0.04, and 0.03 ± 0.01 mJ regarding adhesiveness; 0.39 ± 0.03, 0.46 ± 0.04, and 0.15 ± 0.10 mm regarding springiness; and 1.51 ± 0.14, 1.40 ± 0.09, and 0.99 ± 0.43 mJ regarding chewiness by the 49th day, respectively. Therefore, ENT-NE, CEO-NE, and CO-NE could prevent adverse changes to the texture of salmon fillets, and CO-NE had better potency than ENT-NE and CEO-NE.

#### 3.5.4. Volatile Compound Profile

Thirty-nine compounds were determined in the liquid-smoke salmon fillets, including nine aldehydes, six alcohols, six esters, four phenolic compounds, four hydrocarbons, three ketones, two amines, one acid, and five other compounds (Appendix A). The main aromatic compounds of salmon were identified by screening the volatile substances with an odor activity value (OAV) of ≥1. In this study, thirty-two key aromatic compounds were detected. As shown in Figure 7a, the cumulative variance contribution of the first two principal components was more than 85%, which reflected most of the odor information of the samples. PC1 represents 82.6% of the total variance, and PC2 represents 6.9 % of the total variance. According to Figure 7b, the main aromatic components associated with PC1 were nonanal, hexanal, and octanal. For PC2, phenylethylalcohol, 1-octen-3-ol, 2-methoxy-phenol, and acetic acid provided the main flavor. The characteristic smoky flavor of smoked salmon was captured in PC2. The cluster heatmap was used to reflect the differences in the aromatic components between the different experimental groups (Figure 7c). The results showed that less dimethylamine and trimethylamine were detected in the CO-NE treatment compared to the other groups on the 49th day. 2-methoxy-phenol and 2-Amylfuran were still detected in the CO-NE treatment on the 49th day, while they were barely present in the other three groups. This indicated that a treatment using CO-NE could largely avoid the loss of important flavor components in the fillet samples due to decreased protein degradation.

#### 3.5.5. Sensory Properties

The effect of NEs on the sensory properties of liquid-smoked salmon fillets is shown in Figure 8. Initially (0 days), similar scores were observed for the color, odor, taste, texture, and overall acceptability of the control and treatments, reflecting that the Nes had no adverse effect on the sensory properties of the liquid-smoked salmon fillets. All of the obtained scores of the control were below that of all the smoked salmon treated with Nes from the 7th to the 21st day. Significant differences (*p* < 0.05) were observed between the liquid-smoked salmon fillets in the control and treatments between the 28th–49th day. When compared to the liquid-smoked salmon fillets in the treatments, a slimier texture and increased off-odor and off-color were detected in the fillets in the control samples. Additionally, the largest scores gap was found between the CO-NE treatment and the control. The smoked salmon in the control, ENT-NE, CEO-NE, and CO-NE remained within the acceptability limit (score of 5) until the 28, 35, 35, and 49 days. It was found that CO-NE was the most effective in improving the sensory properties compared to the other treatments. These findings were consistent with the microbiological and physicochemical analysis.

## 4. Discussion

With the growing public demand for safe, natural, and healthy food, the development of various natural antimicrobials for effective food preservatives has attracted more attention. Enterocin Gr17 (ENT) is a bacteriocin that was previously discovered in our laboratory. It has a strong inhibitory activity against a variety of pathogenic bacteria and spoilage bacteria, excellent pH resistance, and heat stability that has the potential to be used as a food preservative [6]. Currently, studies have observed a synergistic inhibitory effect when bacteriocins are combined with EOs to achieve maximum lethality against food-pathogenic and spoilage bacteria. Similar results were also found in the combination of enterocin and EOs, and the greatest combination was ENT and cinnamaldehyde essential oil (CEO). In addition, the antimicrobial efficacy assay in this study was carried out on food pathogens (*S. aureus*, *B. cereus*, *L. monocytogenes*, *S. enterica* and *E. coli*), spoilage bacteria (*P. aeruginosa*, *P. fluorescens*, and *S. putrefaciens*), and pathogenic fungi (*C. albicans*, *A. flavus*, *P. expansum*, and *A. alternata*). Thus far, there have been few studies concerned with the antimicrobial effect on molds and yeast despite the fact that pathogenic fungi can produce more serious and deadly toxins during food storage [38].

The encapsulation of the antimicrobial agents into proper nanoemulsion systems provided a solution to improve the aqueous solubility and physical stability of the EOs and the chemical and thermal stability of the bacteriocins, which may lead to an enhanced antimicrobial effect [14]. However, the difference in the aqueous solubility of bacteriocins (high) and EOs (low) led to a less stable NE system; therefore, the type of emulsifier must be carefully formulated [20]. After the evaluation of droplet size, zeta potential, PDI, TSI, and EE, the NE stabilized with SPI exhibited the best stability, uniformity, and embedding efficiency. Additionally, the NE delivery systems that loaded with ENT/single or mixed were all highly stable. Similarly, the NE containing polylysine and nisin showed the smallest particles and highest stability at a concentration of 1.5% (*w*/*v*) SPI [18]. In addition, a higher DPPH radical-scavenging activity and stronger antimicrobial activities against all of the indicator microorganisms were observed in the CO-NE compared with ENT-NE and CEO-NE, indicating that CO-NE exhibited excellent antimicrobial and antioxidant effects. This was due to the synergy between ENT and CEO and the tiny droplet size of the CO-NE, which made it easier to penetrate the bacteria. Similar results were also found in the co-contained citrullus lanatus seed/phyllanthus niruri methanolic extract and curcumin/resveratrol NE system [39,40]. Then, NE loaded with ENT and CEO was used to delay the spoilage of the liquid-smoked salmon fillet during storage.

Salmon is extremely perishable and would enter lipid oxidation owing to its content of unsaturated fatty acids and prooxidant molecules [1]. The spoilage of salmon is the result of three basic factors: enzymatic autolysis, oxidation, and microbial growth [38]. In our study, liquid-smoked salmon was vacuum-packed and stored at 4 °C to maintain low native enzyme activity. CEO, as a kind of antioxidant, was used to control lipid oxidation, resulting in the development of rancidity and a loss in quality. ENT was added to the smoked salmon to inhibit the microbial growth that was the main reason for the deterioration in salmon quality. Many of the initial microorganisms in salmon do not contribute to spoilage; only specific spoilage organisms (SSOs) were responsible for spoilage. *Pseudomonas* spp. and *Shewanella* spp. were the SSOs for Atlantic salmon (Salmo salar) during low-temperature storage [41]. Accordingly, SSOs participate in spoilage by degrading the nitrogenous compounds in the fish and producing unpleasant and unacceptable off-flavors, causing discoloration or slime [42]. The CO-NE-treated liquid-smoked salmon fillets have the least SSOs among the three NE treatments, which may be caused by the later cumulative antimicrobial effect of lengthening the lag phase of bacterial growth. A similar study also showed that NE-loaded natural bioactive extracts delayed the growth of spoilage microorganisms in fish, extending their shelf life [43].

Following the analysis of the microbial activity, lipid oxidation and proteolysis in liquid-smoked salmon storage were investigated, which were indicated by TBARS and TVB-N. The results suggested that CO-NE exhibited a stronger ability to reduce lipid oxidation and protein degradation in salmon fish than ENT-NE and CEO-NE, which was consistent with the DPPH-scavenging activity of NEs. It was reported that NEs incorporating star anise essential oil, polylysine, and nisin had better antioxidant activity than star anise essential oil-loaded NE for meat products [18]. Interestingly, although CEO-NE had better antimicrobial and antioxidant effects compared to ENT-NE, the difference was not obvious when they were applied to liquid-smoked salmon fillets. This phenomenon revealed the fact that the undesirable influence of complex food systems on the activity of active substances in NEs still exists.

On day 0, no significant differences (*p* > 0.05) were found in the odor, taste, color, and appearance of the NE-treated salmon when compared to the control. It was suggested that the NE delivery system could reduce the impact on the color and flavor of the final food product. As the storage time increased, the *L** values of the CO-NE treatment decreased slower, indicating that CO-NE could decrease the reduction in lightness (*L**) due to several factors, such as protein denaturation, pH change, lipid oxidation, and microbial spoilage. Additionally, the fresh salmon covered with active packaging retained its color compared with the blank packaging samples [44]. Moreover, a slight reduction in *a** was observed in this study, corresponding with the results of color protection using anthocyanidin-compound chitosan nanoparticle edible films on fresh red sea bream fillets [45]. A decrease in the *a** value was observed in all groups with increasing storage time, which was due to the loss of carotenoids and the oxidation of hemoglobin and myoglobin pigments [46]. The decrease in the texture attributes (hardness, adhesiveness, springiness, and chewiness) can be explained by protein degradation due to the effect of endogenous enzymes and microorganisms, breaking the stable structure of the protein [47]. In this study, CO-NE significantly reduced the decrease in the texture attributes during storage between 28th–49th days. The findings of Vital et al. [48] exemplified the effect of the NE with EOs on the texture attributes of fish. A nisin-and-curcumin-loaded nanomat was prepared to enhance the texture of fish in a previous study [49]. 

The results of the volatiles analysis revealed that the CO-NE-treated salmon fillets exhibited an ideal flavor and fewer unpleasant flavors than the control on the 49th day. Similar results were also found in the study concerning the effect of active coatings containing ε-polysine on the volatile compounds of cultured pufferfish [50]. The results of the sensory evaluation showed that CO-NE had a great effect on the color, texture, taste, and odor of salmon fillets, which was reflected by higher scores for each attribute compared to the control. The higher scores in color were probably due to the increased lightness (*L**) and redness (*a**) found in the CO-NE-treated fillets. The higher texture scores could be attributed to the greater elasticity, cohesion, adhesion, and chewiness caused by the protective effect of CO-NE on the lipids and proteins in salmon. The results of the sensory evaluation of odor were consistent with those of the volatile compounds analyses. More positive flavor compounds, such as slimy texture, off-odor, and off-color, were detectable in the CO-NE-treated fillets compared to the control after 49 days of storage. The research related to the application of nanoemulsion based on herb essential oils to rainbow trout fillets also indicated that this nanoemulsion could maintain the sensory parameters (color, texture, taste, and odor) of fish [51]. 

In general, the shelf life of vacuum-packed smoked salmon products was between 14 and 30 days at 4 °C [32]. In our study, the CO-NE-treated liquid-smoked salmon fillets had a longer shelf life of 42 days, and the treatment extended the shelf life by three weeks compared with the control. According to the results of the sensory evaluation, the control, ENT-NE-, CEO-NE-, and CO-NE-treated smoked salmon remained within the acceptability limit (score of 5) until 28, 35, 35, and 49 days, which was consistent with the results of TPC, HBC, and TBARS, while, at the same time, these samples all exceeded the TVC and PBC qualification limit for 7 days. Until 49 days, the TVB-N of the ENT-NE-, CEO-NE-, and CO-NE-treated samples did not exceed the acceptable limit. That was because some SSOs that belong to TPC and HBC play an important role in spoilage, TVC and PBC also include many microorganisms that do not contribute to spoilage. Additionally, the deterioration of sensory properties is caused by spoilage owing to microbial growth and TVC and PBC rapidly increased during the initial stage of fillets storage. The different results between TBARS and TVB-N may be due to the decomposition of protein, which is a slight but continuous process. For cold-smoked salmon, the variation in the microorganism’s population over the limit value is slightly earlier than the signs of spoilage indicated by sensory changes [52,53]. The high salt cold-smoked salmon stored at 5 and 10 °C were acceptable for at least 2–3 weeks, while the numbers of TVC and PBC reached a level of 10^7^–10^8^ CFU/g after 12 days of storage [52]. Meanwhile, smoked salmon stored at 2 °C was acceptable for 6 weeks, while the PBC for smoked salmon exceeded the value of 10^6^ CFU/g on 5 weeks in cold storage conditions [53].

## 5. Conclusions 

In this study, a nanoemulsion system incorporating enterocin Gr17 and cinnamaldehyde essential oil (CEO) was successfully developed and characterized for application to liquid-smoked salmon fillets storage. The nanoemulsion system incorporating enterocin Gr17 and CEO (CO-NE) could significantly inhibit the growth of microflora, suppress the accumulation of TVB-N and TBARS, and maintain better color, texture, and sensory profiles during smoked salmon storage at 4 °C. From a microbiological, physicochemical, and sensory point of view, the CO-NE treatment could extend the shelf life to 42 days. In summary, the present work demonstrates that the combination of enterocin Gr17 and CEO could be a promising bio-preservative technology and an alternative to the conventional processes used for improving the safety of fish products at lower additive concentrations of both natural bioactive agents. In our laboratory, a detailed study on the synergetic mechanism between enterocin Gr17 and CEO on the inactivation of foodborne pathogens is currently underway.

## Figures and Tables

**Figure 1 foods-12-00078-f001:**
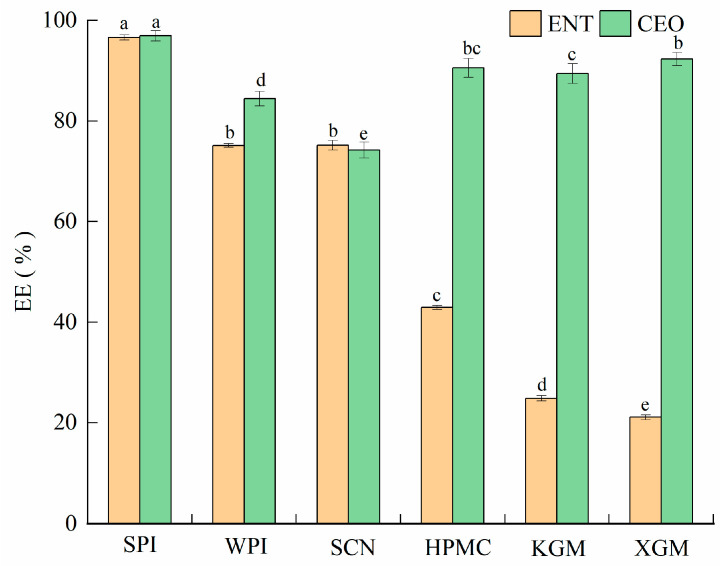
The encapsulation efficiency (EE) of enterocin Gr17 (ENT)/cinnamaldehyde (CEO) in nanoemulsions produced by using soybean protein isolate (SPI), whey protein isolate (WPI), sodium caseinate (SCN), hydroxypropyl methyl-cellulose (HPMC), konjac gum (KGM) and xanthan gum (XGM). Means with no letter in common are significantly different (*p* < 0.05).

**Figure 2 foods-12-00078-f002:**
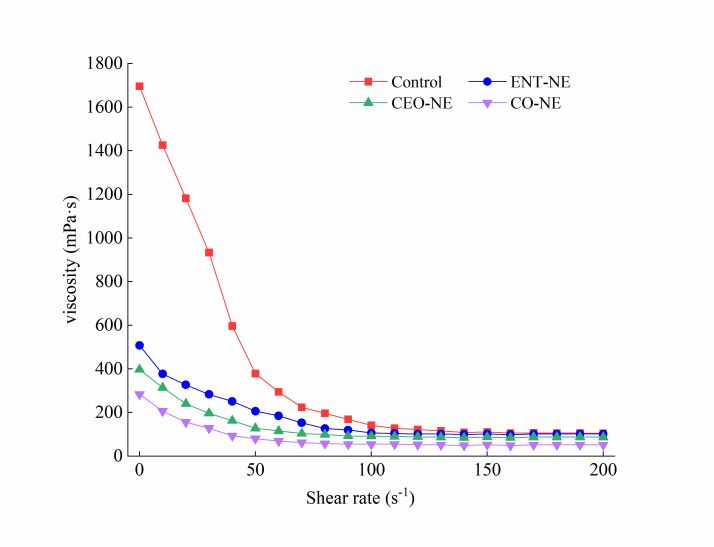
The viscosity of nanoemulsions containing enterocin Gr17 alone (ENT-NE), cinnamaldehyde alone (CEO-NE), enterocin Gr17 and cinnamaldehyde (CO-NE) and no enterocin Gr17/cinnamaldehyde (Control).

**Figure 3 foods-12-00078-f003:**
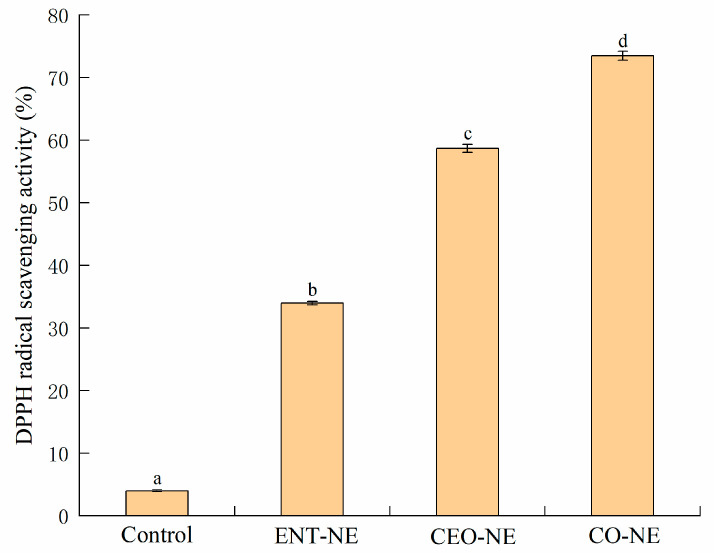
The DPPH radical scavenging activity of nanoemulsions containing enterocin Gr17 alone (ENT-NE), cinnamaldehyde alone (CEO-NE), enterocin Gr17 and cinnamaldehyde (CO-NE) and no enterocin Gr17/cinnamaldehyde (Control). Bars with no letter in common for each treatment are significantly different (*p* < 0.05).

**Figure 4 foods-12-00078-f004:**
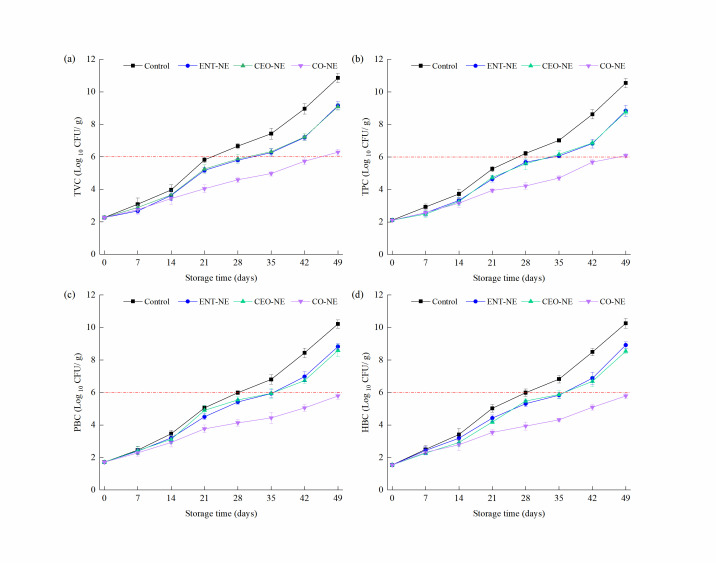
Changes of total viable counts (TVC) (**a**), total *Pseudomonas* bacteria counts (TPC) (**b**), psychrotrophic bacteria counts (PBC) (**c**) and hydrogen sulfide-producing bacteria counts (HBC) (**d**) of liquid-smoked salmon fillets treated with nanoemulsions containing enterocin Gr17 alone (ENT-NE), cinnamaldehyde alone (CEO-NE), enterocin Gr17 and cinnamaldehyde (CO-NE) and no enterocin Gr17/cinnamaldehyde (Control) during storage at 4 °C for 49 days. The red dotted line means the minimal quantification limit.

**Figure 5 foods-12-00078-f005:**
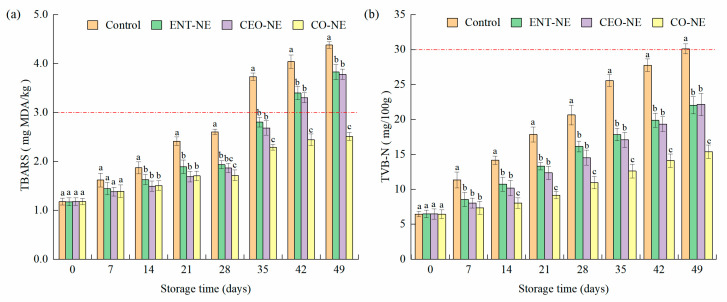
Changes in TBARS (**a**) and TVB-N (**b**) values of liquid-smoked salmon fillets treated with nanoemulsions containing enterocin Gr17 alone (ENT-NE), cinnamaldehyde alone (CEO-NE), enterocin Gr17 and cinnamaldehyde (CO-NE) and no enterocin Gr17/cinnamaldehyde (Control) during storage at 4 °C for 49 days. The red dotted line means the minimal acceptable limit. Bars with no letter in common for each treatment are significantly different (*p* < 0.05).

**Figure 6 foods-12-00078-f006:**
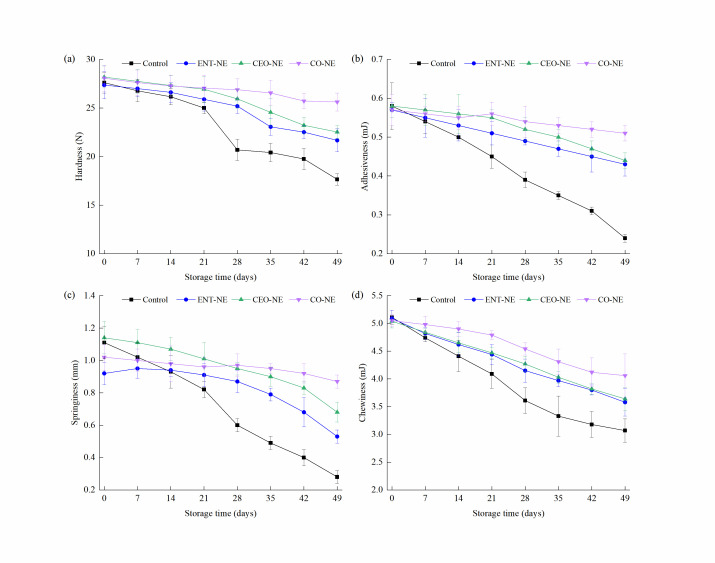
Changes in hardness (**a**), adhesiveness (**b**), springiness (**c**) and chewiness (**d**) of liquid-smoked salmon fillets treated with nanoemulsions containing enterocin Gr17 alone (ENT-NE), cinnamaldehyde alone (CEO-NE), enterocin Gr17 and cinnamaldehyde (CO-NE) and no enterocin Gr17/cinnamaldehyde (Control) during storage at 4 °C for 49 days.

**Figure 7 foods-12-00078-f007:**
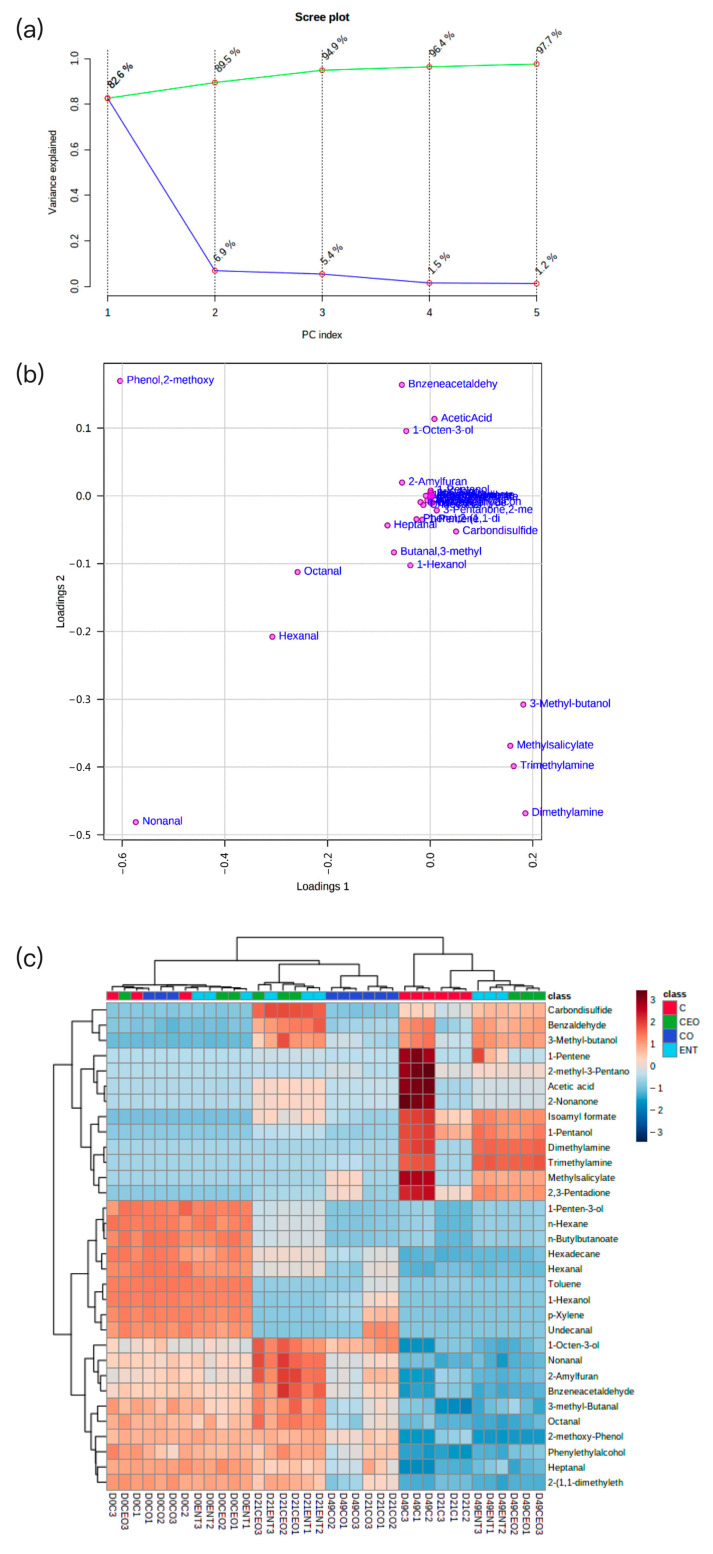
Principal component analysis (**a**), principal components contribute to variance (**b**) and heat-map analysis (**c**) based on volatile-compound profile of liquid-smoked salmon fillets treated with nanoemulsions containing enterocin Gr17 alone (ENT-NE), cinnamaldehyde alone (CEO-NE), enterocin Gr17 and cinnamaldehyde (CO-NE) and no enterocin Gr17/cinnamaldehyde (Control) during storage at 4 °C on 0th, 21st, 49th day.

**Figure 8 foods-12-00078-f008:**
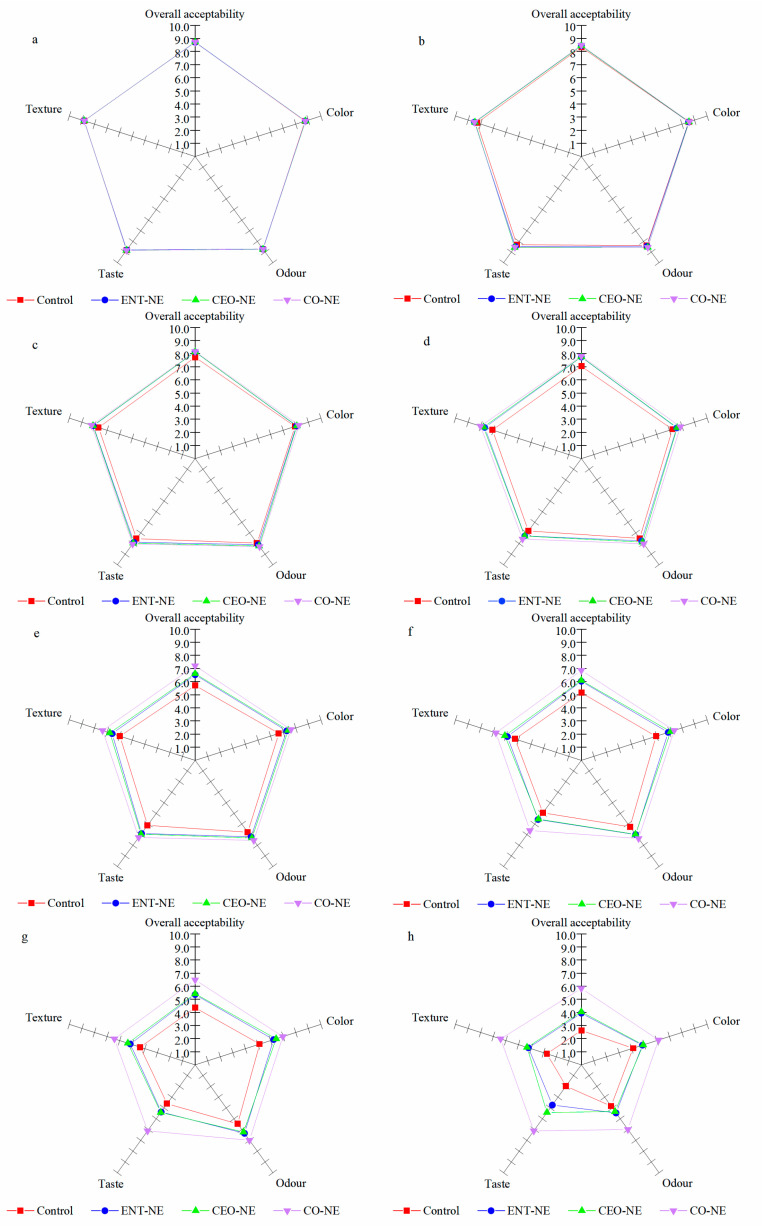
Sensory characteristics of liquid-smoked salmon fillets treated with nanoemulsions containing enterocin Gr17 alone (ENT-NE), cinnamaldehyde alone (CEO-NE), enterocin Gr17 and cinnamaldehyde (CO-NE) and no enterocin Gr17/cinnamaldehyde (Control) during storage at 4 °C on 0th (**a**), 7th (**b**), 14th (**c**), 21st (**d**), 28th (**e**), 35th (**f**), 42nd (**g**), 49th (**h**) day. The acceptable limit value of sensory properties is 5 for liquid-smoked salmon.

**Table 1 foods-12-00078-t001:** The minimum inhibitory concentration (MIC) and fractional inhibitory concentration index (FICI) of enterocin Gr17 and essential oils (eugenol [EEO], thymol [TEO], carvacrol [CAEO], cinnamaldehyde [CEO], menthone [MEO] and linalool [LEO]) against bacterial and fungal strains.

Indicator Strains	MIC (g/L)							FICI						
	ENT	EEO	TEO	CAEO	CEO	MEO	LEO	ENT and EEO	ENT and TEO	ENT and CAEO	ENT and CEO	ENT and MEO	ENT and LEO	
*S. aureus*	5	2.5	5	2.5	0.3125	1.25	1.25	1.031	0.531	0.563	0.125	1.25	1.031	
*B. cereus*	1.25	2.5	5	1.25	0.3125	1.25	0.625	0.125	0.156	1.5	0.063	0. 5	1.25	
*L. monocytogenes*	1.25	1.25	0.3125	0.3125	0.1563	0.625	0.3125	0.625	1.125	0.156	0.094	0.563	0.625	
*S. enterica*	1.25	0.3125	5	0.1563	0.3125	2.5	0.3125	1.125	1.25	0.625	0.125	0.75	1.063	
*E. coli*	5	2.5	5	0.3125	0.3125	2.5	0.625	0.125	0.563	0.75	0.063	0.625	1.5	
*P. aeruginosa*	5	1.25	0.625	0.3125	0.3125	5	0.625	1.5	0.625	1.031	0.125	1.125	0.563	
*P. fluorescens*	5	1.25	0.625	0.3125	0.3125	5	5	0.375	1.25	0.188	0.125	0.75	0.75	
*S. putrefaciens*	5	1.25	0.625	0.3125	0.3125	5	2.5	2	0.75	0.625	0.125	1.25	1.25	
*C. albicans*	10	0.3125	0.1563	0.0781	0.0195	0.1563	0.0781	1.5	0.563	1.031	0.188	2	2	
*A. flavus*	ND	0.3125	0.3125	0.1563	0.0391	0.1563	0.1563	1.25	1.031	1.5	0.125	1.063	0.75	
*P. expansum*	ND	0.625	0.625	0.3125	0.3125	0.3125	0.1563	1.063	1.5	2	0.125	1.031	1.5	
*A. alternata*	20	0.3125	0.625	0.1563	0.0781	0.625	0.625	1.031	1.5	0.75	0.188	1.25	1.5	

Abbreviations: MIC, minimum inhibitory concentration; FICI, fractional inhibitory concentration index; ENT, enterocin Gr17; EEO, eugenol essential oil; TEO, thymol essential oil; CAEO, carvacrol essential oil; CEO, cinnamaldehyde essential oil; MEO, menthone essential oil; LEO, linalool essential oil. ND: not determined.

**Table 2 foods-12-00078-t002:** The droplet size, zeta potential, polydispersity index (PDI) and Turbiscan stability index (TSI) of nanoemulsions containing enterocin Gr17 and cinnamaldehyde produced by using soybean protein isolate (SPI), whey protein isolate (WPI), sodium caseinate (SCN), hydroxypropyl methyl-cellulose (HPMC), konjac gum (KGM) and xanthan gum (XGM).

Emulsifier	Average Droplet Size (nm)	Zeta Potential (mV)	PDI	TSI (12 h)
SPI	161.26 ± 6.40 ^f^	−32.51 ± 0.83 ^d^	0.235	2.50 ± 0.06 ^f^
WPI	219.23 ± 6.93 ^e^	−21.41 ± 0.49 ^b^	0.283	5.56 ± 0.07 ^e^
SCN	344.62 ± 8.08 ^d^	−27.60 ± 0.48 ^c^	0.241	6.79 ± 0.09 ^d^
HPMC	609.13 ± 7.11 ^b^	−12.41 ± 0.38 ^a^	0.689	12.45 ± 0.52 ^a^
KGM	372.94 ± 8.53 ^c^	−13.23 ± 0.21 ^a^	0.501	11.03 ± 0.41 ^b^
XGM	1215.00 ± 8.89 ^a^	−43.64 ± 0.98 ^e^	0.747	8.43 ± 0.40 ^c^

Abbreviations: PDI, polydispersity index; TSI, Turbiscan stability index; SPI, soybean protein isolate; WPI, whey protein isolate; SCN, sodium caseinate; HPMC, hydroxypropyl methylcellulose; KGM, konjac gum; XGM, xanthan gum. Values are expressed as means ± standard deviation. Columns with no lowercase letter in common for each treatment are significantly different (*p* < 0.05).

**Table 3 foods-12-00078-t003:** The antimicrobial activity of nanoemulsions containing enterocin Gr17 alone (ENT-NE), cinnamaldehyde alone (CEO-NE), enterocin Gr17 and cinnamaldehyde (CO-NE) and no enterocin Gr17/cinnamaldehyde (Control).

Samples	Inhibitory Zone (mm)											
	*S. aureus*	*B. cereus*	*L. monocytogenes*	*S. enterica*	*E. coli*	*P. aeruginosa*	*P. fluorescens*	*S. putrefaciens*	*C. albicans*	*A. flavus*	*P. expansum*	*A. alternata*
Control	ND	ND	ND	ND	ND	ND	ND	ND	ND	ND	ND	ND
CEO-NE	20.7 ± 0.65 ^b^	17.0 ± 0.59 ^b^	18.77 ± 0.55 ^b^	15.87 ± 0.64 ^b^	14.90 ± 0.46 ^b^	18.97 ± 0.93 ^b^	12.30 ± 0.44 ^b^	11.80 ± 0.40 ^b^	26.13 ± 0.25 ^b^	21.7 ± 0.65 ^b^	15.13 ± 0.83 ^a^	16.97 ± 0.45 ^b^
ENT-NE	12.8 ± 0.26 ^c^	13.7 ± 0.45 ^c^	12.43 ± 0.15 ^c^	13.13 ± 0.25 ^c^	11.80 ± 0.26 ^c^	10.87 ± 0.42 ^c^	12.07 ± 0.29 ^b^	11.67 ± 0.50 ^b^	11.73 ± 0.49 ^c^	ND	ND	9.90 ± 0.44 ^c^
CO-NE	42.4 ± 0.31 ^a^	34.6 ± 0.66 ^a^	30.63 ± 0.80 ^a^	33.67 ± 0.35 ^a^	25.17 ± 0.76 ^a^	30.83 ± 0.45 ^a^	25.40 ± 0.40 ^a^	21.83 ± 0.96 ^a^	38.93 ± 0.78 ^a^	22.2 ± 0.72 ^a^	16.33 ± 0.90 ^a^	30.30 ± 0.45 ^a^

Values are expressed as means ± standard deviation. Columns with no lowercase letter in common for each treatment are significantly different (*p* < 0.05). ND: not determined.

**Table 4 foods-12-00078-t004:** Changes on instrumental color parameters (CIE *L**, *a**, *b**) of liquid-smoked salmon fillets treated with nanoemulsions containing enterocin Gr17 alone (ENT-NE), cinnamaldehyde alone (CEO-NE), enterocin Gr17 and cinnamaldehyde (CO-NE) and no enterocin Gr17/cinnamaldehyde (Control) during storage at 4 °C for 49 days.

	Storage Time (days)	Control	ENT-NE	CEO-NE	CO-NE
*L*(Lightness)*	0	53.15 ± 2.07 ^Aa^	53.83 ± 1.15 ^Aa^	54.62 ± 0.63 ^Aa^	53.93 ± 0.99 ^Aa^
	7	53.84 ± 1.25 ^Aa^	54.40 ± 0.49 ^Aa^	54.19 ± 1.04 ^Aa^	53.52 ± 1.25 ^Aa^
	14	53.67 ± 0.72 ^Aa^	54.15 ± 1.10 ^Aa^	54.14 ± 0.40 ^Aa^	53.39 ± 0.34 ^Aa^
	21	52.85 ± 0.81 ^Aba^	52.47 ± 0.47 ^Aba^	53.74 ± 0.81 ^Aba^	53.45 ± 1.69 ^Aa^
	28	50.51 ± 0.46 ^Bb^	51.83 ± 0.57 ^Bab^	52.61 ± 0.97 ^Bca^	53.51 ± 1.31 ^Aa^
	35	44.07 ± 1.82 ^Cb^	51.38 ± 0.61 ^Ba^	51.88 ± 0.25 ^Ca^	53.17 ± 1.85 ^Aa^
	42	40.57 ± 1.38 ^Dd^	47.59 ± 0.76 ^Cc^	50.00 ± 0.84 ^Db^	53.03 ± 0.75 ^Aa^
	49	37.64 ± 1.70 ^Ec^	45.40 ± 0.88 ^Db^	46.34 ± 0.80 ^Eb^	52.86 ± 1.64 ^Aa^
*a*(redness)*	0	22.06 ± 1.16 ^Aa^	22.09 ± 0.90 ^Aa^	22.14 ± 1.46 ^Aa^	21.78 ± 0.90 ^Aa^
	7	20.45 ± 0.99 ^Aba^	21.99 ± 0.98 ^Aa^	22.24 ± 0.87 ^Aa^	21.89 ± 1.67 ^Aa^
	14	19.22 ± 1.40 ^Ba^	21.65 ± 0.72 ^Aa^	21.29 ± 1.02 ^Aa^	20.76 ± 1.01 ^Aa^
	21	19.79 ± 0.94 ^Bb^	20.91 ± 0.65 ^Aab^	21.44 ± 0.71 ^Aa^	20.88 ± 0.59 ^Aa^
	28	16.64 ± 0.49 ^Cb^	21.13 ± 1.08 ^Aa^	22.26 ± 0.85 ^Aa^	22.11 ± 1.05 ^Aa^
	35	15.64 ± 0.72 ^Cc^	19.26 ± 1.12 ^BCb^	19.80 ± 1.02 ^Aab^	22.04 ± 2.27 ^Aa^
	42	13.76 ± 1.30 ^Db^	17.70 ± 1.63 ^Ca^	17.70 ± 1.33 ^Ba^	20.19 ± 0.64 ^Aa^
	49	11.65 ± 0.95 ^Ec^	15.54 ± 1.96 ^Db^	17.04 ± 1.72 ^Bb^	21.14 ± 1.28 ^Aa^
*b*(yellowness)*	0	28.28 ± 1.55 ^Ba^	27.35 ± 0.97 ^Aa^	28.30 ± 1.20 ^Aa^	27.92 ± 0.66 ^Aa^
	7	28.18 ± 0.87 ^Ba^	27.75 ± 1.09 ^Aa^	28.11 ± 0.96 ^Aa^	28.09 ± 0.91 ^Aa^
	14	28.35 ± 0.60 ^Ba^	28.14 ± 0.51 ^Aa^	28.24 ± 1.66 ^Aa^	28.13 ± 1.54 ^Aa^
	21	28.57 ± 0.51 ^Ba^	28.45 ± 2.06 ^Aa^	28.54 ± 1.50 ^Aa^	28.09 ± 1.06 ^Aa^
	28	29.08 ± 1.88 ^Aba^	28.60 ± 0.69 ^Aa^	28.45 ± 1.67 ^Aa^	28.23 ± 1.72 ^Aa^
	35	30.08 ± 1.49 ^Aba^	29.03 ± 1.60 ^Aa^	29.04 ± 1.95 ^Aa^	28.27 ± 1.18 ^Aa^
	42	31.41 ± 1.70 ^Aa^	29.27 ± 1.86 ^Aa^	29.28 ± 1.53 ^Aa^	28.32 ± 1.91 ^Aa^
	49	30.60 ± 1.82 ^Aa^	29.63 ± 1.93 ^Aa^	29.51 ± 1.39 ^Aa^	28.55 ± 1.26 ^Aa^

Values are expressed as means ± standard deviation. Columns with no uppercase letters in common for each storage time are significantly different and rows with no lowercase letter in common for each treatment are significantly different (*p* < 0.05).

## Data Availability

Data is contained within the article.

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
