# Peer review of "Effect of Nanoemulsion Containing Enterocin GR17 and Cinnamaldehyde on Microbiological, Physicochemical and Sensory Properties and Shelf Life of Liquid-Smoked Salmon Fillets"

_foods, 2022, doi:10.3390/foods12010078_

Round 1
Reviewer 1 Report
Dear authors
Please, find the attached file.

Author Response
Thanks very much for your comments concerning our manuscript. Those comments are all valuable and very helpful for revising and improving our paper. We have studied comments carefully and have made corrections accordingly which we hope meet with your approval. Please see the attachment for detailed responses.

Reviewer 2 Report
Dear Editor,
Manuscript Foods-2070164 reports on the effect of nanoemulsion containing enterocin Gr17 and cinnamaldehyde on microbiological, physico-chemical and sensory quality characteristics of smoked salmon fillets during storage . In my opinion, this is a study has several flaws which should be seriously taken into consideration. The main flaw is that based on values recorded for microbiological, physico-chemical and sensory parameters the shelf life of CO-NE treated smoked salmon is substantially more than 21-35 days. Even if this was correct, this very large range of values between day 21 and 35 is not very informative as there is a very large difference between 3 (21 days) and 5 weeks (35 days). Furthermore in Table 1 the authors present data on MIC and FICI on parameters i.e. EEO, TEO, CAEO, MEO and LEO which were never mentioned in the Materials and methods section. A third problem is that English is very poor and the authors should have consulted a native English speaker before submitting their work. It is not my job to correct the English language in a technical text.
My detailed comments follow the text sequence:
The title should be changed to include the microbiological, physico-chemical and sensory properties and shelf life of smoked salmon fillets.
l.26: the two weeks’ shelf life extension does not agree with the suggested product (CO-NE treatment) shelf life of 21-35 days mentioned on l.567. At this point I should mention that the ultimate criterion for the determination of a products’ shelf life is sensory analysis, excluding of course the presence of pathogenic microorganisms such as Clostridium botulinum type E. In the present study data shows that the CO-NE treatment resulted in an acceptable product until day 49 of storage ! relevant to this issue is the fact that the authors never mention the sensory acceptable upper limit score. Was this 5 or 6 or whatever.
l.39: the protein and fat content of smoked salmon should be included in the text.
l.58: change ¾ to percent i.e. 75 %
l.78: was Enterocin GR17 discovered by the authors ?
l.193: change ‘property’ to ‘parameters’
l.209: change ‘moves’ to ‘operated at’
l.212: change ‘microbial’ to ‘Microbiological’
l.220: change ‘flavor’ to ‘Volatile’
Section 2.7.4.: the authors should briefly describe the method of volatile compound determination. I,e, which fiber was used, how was this fiber chosen and which were the experimental conditions during the SPME method application i.e. equilibration time, adsorption time, chromatographic temperature program used, etc.
l.232: was this method a descriptive or affective (acceptability) sensory method ? Also what was the scoring scale and lower limit of product acceptability ?
Section 2.8: if analytical variables were treatment and time, I believe that the authors should have used a two-way and not one-way analysis.
Table 1: parameters EEO, TEO, CAEO MEO and LEO were never mentioned in the Materials and methods section ! Also, foot notes should explain all abbreviations shown in the Table. Also, +/- std. deviation plus superscripts are missing from the Table.
Table 2: PDI and TSI abbreviations should be given in full as a foot note.
l.329: change “microbial properties’ to ‘Microbiological parameters’
l.339: the upper acceptable limit for smoked salmon of the ICMSF is 106-107 log cfu/g Thus, the limit of 105 log cfu/g mentioned is questionable. If this limit change to 107 log cfu/g, then all the discussion on the microbiological quality of the product
change !
Fig. 5a: the authors should refer to the literature regarding the upper limit for TBARS resulting in product off-odors.
Fig. 5b: the authors should refer to the literature regarding the upper limit for TVBN of 30-35 mg/100g. Even with an upper limit of 25 mg/100 g the CO-NE treatment scored 15-17 mg/100 g on day 49 which is considered acceptable. Thus, the suggested shelf life should be higher based on the combination of physico-chemical, microbiological and sensory parameter values.
l.439: change ‘Flavor substance’ to ‘’Volatile compound profile’
Fig. 7: see my general comments on the sensory shelf life of the product.
l.539: change ‘smell’ to ‘odor’
l.557: a comparison of microbiological, physicochemical and sensory attributes should be included in the text showing agreement or disagreement among them.
l.567: see my general comments on the actual shelf life of the CO-NH treated product.
Author Response

(The authors gave the same response as above.)

Round 2
Reviewer 1 Report
Dear authors,
Please, find the attached file.

Author Response
Thanks very much for your comments concerning our manuscript. These comments are all valuable and very helpful for revising and improving our paper, as well as the important guiding significance to our researches. We have studied comments carefully and have made correction which we hope meet with approval. Revised portion are highlighted in yellow. Please see the attachment for detailed responses.
